# Factors Associated with HPV Genital Warts: A Self-Reported Cross-Sectional Study among Students and Staff of a Northern University in Nigeria

**DOI:** 10.3390/v16060902

**Published:** 2024-06-02

**Authors:** Melvin Omone Ogbolu, Olanrewaju D. Eniade, Hussaini Majiya, Miklós Kozlovszky

**Affiliations:** 1BioTech Research Center, University Research and Innovation Center, Óbuda University, Bécsi Street 96/B, 1034 Budapest, Hungary; 2Department of Epidemiology and Medical Statistics, College of Medicine, University of Ibadan, CW22+H4W, Queen Elizabeth II Road, Agodi, Ibadan 200285, Nigeria; olanrewaju.eniade@ifain.org; 3International Foundation against Infectious Disease in Nigeria (IFAIN), 6A, Dutse Street, War College Estate, Gwarimpa, Abuja 900108, Nigeria; 4Department of Microbiology, Ibrahim Badamasi Babangida University, 3H89+XW3, Minna Road, Lapai 911101, Nigeria; hussainimajiya@ibbu.edu.ng; 5John von Neumann Faculty of Informatics, Óbuda University, Bécsi Street 96/B, 1034 Budapest, Hungary; kozlovszky.miklos@nik.uni-obuda.hu; 6Medical Device Research Group, LPDS, Institute for Computer Science and Control (SZTAKI), Hungarian Research Network (HUN-REN), 1111 Budapest, Hungary

**Keywords:** HPV awareness programs, HPV infection, HPV genotypes, HPV genital warts, HPV knowledge, HPV vaccine

## Abstract

The menace of human papillomavirus (HPV) infections among low- and middle-income countries with no access to a free HPV vaccine is a public health concern. HPV is one of the most common sexually transmitted infections (STIs) in Nigeria, while the most known types of HPV genotypes being transmitted are the high-risk HPV-16 and 18 genotypes. In this study, we explored the predictors of self-reported HPV infections and HPV genital warts infection among a population of students, non-academic staff, and academic staff of Ibrahim Badamasi Babangida (IBB) University located in Lapai, Nigeria. We also assessed their knowledge about HPV infections and genotypes, and sexual behaviors. An online cross-sectional study was conducted by setting up a structured questionnaire on Google Forms and it was distributed to the university community via Facebook and other social media platforms of the university. The form captured questions on HPV infection, and knowledge about HPV infection and genotypes, as well as the sexual health of the participants. All variables were described using frequencies and percentage distribution; chi-squared test statistics were used to explore the association between HPV infection (medical records of HPV infection) and the participants’ profile, and a logistic regression analysis was performed to examine the factors associated with HPV genital warts infection among the population. This study reveals those participants between the ages of 26–40 years (81.3%) and those currently not in a sexually active relationship—single/divorced (26.4%)—who have self-reported having the HPV-16 and -18 genotypes. Moreover, participants between 26–40 years of age (OR: 0.45, 95%CI: 0.22–0.89) reported themselves to be carriers of HPV genital warts. Therefore, this study reveals the factors associated with HPV infection and genital warts peculiar to IBB university students and staff. Hence, we suggest the need for HPV awareness programs and free HPV vaccine availability at IBB university.

## 1. Introduction

Human papillomavirus (HPV) is one of the most common sexually transmitted infections (STIs) among adults, of which HPV 16, 18, 31, and 35 (high-risk types) [1] are the most prevalent HPV genotypes among sexually active adults in the northern part of Nigeria [2,3]. These high-risk HPV genotypes, especially HPV 16 and 18, have been clinically proven to be the leading cause of cervical cancer in women across low-income African countries [4,5]. However, the low-risk (11, 42, 61, 70, and 81) HPV genotypes are capable of causing HPV genital warts [6]. Numerous studies have revealed that HPV is also highly associated with the growth of HPV genital warts in both men and women [7,8]. In most cases, persons with a healthy immune system can be HPV-free over time even after being infected. An HPV vaccine-built immunity and other types of preventative vaccines (such as toxoids, messenger ribonucleic acid (mRNA), and subunit vaccines [9]), alongside with some other factors (such as healthy weight, healthy food, exercise, and staying away from excessive alcohol intake and smoking [10]) can effectively repel against persistent HPV infections within two years before it develops into a tumor [11] and other immunological diseases [12] in the presence of a healthy immune system. However, north-central Nigerian adults are faced with some challenges which have contributed to the spread of HPV infection such as the unavailability of some protective measures. Although this is associated with the lack of knowledge of HPV in general, the following are also factors: abject poverty (resulting in the inability to purchase HPV vaccines, eat healthy food, and attend paid HPV-related seminars), individuals’ ethical imperatives, the unavailability of free medical services, and some national policies. In terms of national policies, this is a contributing factor such that the government does not put in place policies to make an HPV vaccine available at a low cost and enforce it as a part of the routine vaccination program in Nigeria [13]. Due to these challenges, there have been increases in the cases of HPV infection and the cervical cancer mortality rate among women in the north-central part of Nigeria yearly [14]. According to the statistical report gathered by the ICO/IARC Information Centre on HPV and Cancer (HPV Information Centre) in October 2021, among other kinds of cancers, the HPV-related (HPV 16 and 18) cancer occurrence in the female population of ages ≥ 15 years in Nigeria was 66.9% amidst 56.2 million total females who are at risk of cervical cancer [15].

Some epidemiological studies have also in the past revealed demographic variations among low-income settlements in Nigeria and its association with HPV infection [6]. For instance, a study conducted for 16 years in a rural area in Nigeria reveals that inaccessibility to childhood and adolescent routine immunization as a preventive measure for preventable diseases [16] is more prevalent in low- and middle-income areas in Nigeria and other sub-Saharan African (SSA) countries, causing severe infant mortality [17], thereby putting non-HPV vaccinated adults at risk of persistent HPV infections. Similarly, a study conducted among adolescent and young girls in Jos, Nigeria also shows that the lack of HPV vaccine and lack of knowledge about HPV enhanced the prevalence of HPV among the population [6]. Hence, it has been proven that the availability of free medical services to provide organized HPV programs, HPV vaccines, and HPV testing and cervical cancer screenings using cytology/visual inspection/colposcopic biopsy in low-income settlements would help to mitigate HPV infection cases and the cervical cancer mortality rate in Nigeria [2,5,18].

In this study, we explored the demographic predictors of self-reported [19] HPV infections and HPV genital warts infection among low-income students, and non-academic, and academic (lecturers) staff of Ibrahim Badamasi Babangida (IBB) University located in Lapai, Nigeria, considering their income rate as compared to the country’s minimum wage and if they have reported to have had an HPV infection in the past. Furthermore, age as another demographic factor was also analyzed to observe if the population’s age is likely to affect their monthly income rate. Abject poverty as one of the predictors of HPV infection was considered for analysis in this study, especially in low-income and lower-middle-income countries (LMICs) (as in Nigeria) [20], where the citizens cannot fend for their medical services in the presence of a Gross National Income (GNI) per capita at $2157 as recorded in 2019 by the World Bank collection of development indicators.

## 2. Motivation—Problem Definition

In the absence of knowledge about HPV and cervical cancer even in an academic/educational environment, there is a likelihood that there would be cases of HPV infection among a population. Sadly, the male population in the society are not aware of their HPV status, thereby causing females to harbor the risk of contracting HPV, which further develops into cervical cancer in the future [21]. Clinical research has revealed that HPV takes 10–15 years to develop into abnormal cells [22]. The purpose of this paper is to reveal the risk factors of HPV infection and genital warts in an academic environment and study those who have self-reported cases of HPV infection and genital warts using statistical methods. Our results can be adopted by the Niger State Ministry of Health for HPV awareness in an academic setting, healthcare sectors, and private organizations, and by individuals as a guide for safe sexual practices and knowledge about HPV.

## 3. Methods and Materials

### 3.1. Study Design and Setting

This was an online cross-sectional study among students, and non-academic and academic staff of Ibrahim Badamasi Babangida (IBB) University, Lapai, Niger State, Nigeria. IBB University comprises students and staff from various regions in Nigeria, which includes Igbo, Yoruba, and Hausa major ethnic groups. However, data to determine which regions the participants are from were not captured in our questionnaire. Taking into consideration the privacy of the participants, Nigerian citizens have right to their private data and the Nigerian constitution guarantees privacy protection to all citizens. Hence, the Nigeria Data Protection Regulation (NDPR) was enforced during data collection [23].

#### 3.1.1. Sample Size

The sample size determination formula for a cross-sectional study was used in this study:(1)$$n=\frac{Z_{\alpha }^{2}\times pq\times d_{eff}}{d^{2}}$$
where:

*n* = minimum sample size;

*z* = standard normal deviation of 95% confidence level—corresponds to a value of 1.96;

*p* = record of HPV genital warts from similar studies in southwestern Nigeria was (0.22) [24];

*e* = level of precision of 0.04;

*q* = 1 − *p*; 1 − 0.22 = 0.78;

*d_eff_* = 1.98—a design effect (*d_eff_*) of 1.98 was factored because the study cut across clusters (i.e., cluster of students, and non-academic and academic staff);

*n* = 1615;

*n_a_* (after adjusting for a 10% non-response rate) = 1860.

A total of 1860 eligible people who completed the online questionnaires were considered for/enrolled in this study (Figure 1).

#### 3.1.2. Sampling Method

Having established a high incidence of sexual behavior [25] which is a risk factor for HPV infection among university students and staff [26,27]. we, thus, carried out this study among the university community, at Ibrahim Badamasi Babangida University, Lapai in Niger State, Nigeria. IBB University is located in the north-central region of Nigeria and has a student capacity of about 7000, 820 non-academic staff, and 310 academic staff. The data were collected using an online data collection tool (Google survey form). The questionnaire was set up on the Google survey form and the function that restricts multiple responses from participants was enabled.

#### 3.1.3. Study Instrument and Data Collection

A structured questionnaire was used, which captured the demographic characteristics, information about HPV and sexual health, and knowledge about early sex and HPV-16 and -18 genotype. In order to improve the response rate, we raised awareness (live event) about the study through the Facebook platform during the COVID-19 pandemic in 2019, but data collection lasted between 2019–2022. The lead investigator and a group of gynecologists in Nigeria and the United States of America facilitated the event. During the event, participants were asked to fill in an online questionnaire—the Human Papillomavirus (HPV) and Cervical Cancer Risk Assessment (HCRA) Tool (also known as the HPV Assessment Test (HAT) Tool [1]). The link to the online questionnaire (Google Forms) was distributed via Facebook and email address (it was sent to students who could not attend the online event). Google Forms is an electronic platform designed for creating surveys, quizzes, and knowledge evaluations. The Google Form can be created and administered on personal computers, and mobile devices, as well as tablets. Some of its important features are real-time data capturing, friendly user interface, the ability to limit responses to once per person, etc. The Google Form has proven to be effective for research data collection and other data-capturing activities [28].

#### 3.1.4. Study Variables

The outcome variable for this study was self-reported medical record/history of HPV genital warts (Yes or No). The explanatory variables included the 1. Demographic variables: Gender [Male, Female]; Age group (15–25, 26–40, 41–64); Level of education [High school/equivalent, and BSc/MSc/Doctorate degree)]; Income in USD ($) [Below 100, 100–200, above 200], $100 ≡ #45,000 Nigerian currency; and Marital Status [Married—currently in a sexually active relationship, not in a sexually active relationship (single/divorced)]; 2. Sexual Health/Behavior: Sexual intercourse [Yes, No]; Age at first sex (15–20, 21–30, >30 years); Number of the sexual partner in a lifetime [1, 2–4, >4]; Kind of sex [vaginal only, others (anal, oral, and vaginal)]; Knowledge about early sex [Yes, No]; and Protected sex, defined as the use of condom for sexual intercourse [unprotected, protected]; and 3. Information on HPV factors: Vaccinated against HPV [Yes, No]; HPV seminar attendance [Yes, No]; and Knowledge of HPV-16 and -18 genotypes [Yes, No].

### 3.2. Data Analysis

We summarized all the variables using frequencies and percentages (descriptive statistics). Moreover, a chi-square test of association was carried out to test the association between the participants’ profiles and self-reported medical record/history of HPV genital warts. We examined the factors associated with HPV infection by fitting a binary logistic regression, and an adjusted odds ratio (_Adj_OR) was reported.

Model Expression

The model expression implemented for statistical analysis is the logistic regression model [7]:(2)$$Log\frac{p_{i}}{1-p_{i}}=\beta_{0}+X_{1}\beta_{1}+X_{2}\beta_{2}+X_{3}\beta_{3}+\varepsilon_{i}$$
where:

Log $\frac{p_{i}}{1-p_{i}}$ is the probability of HPV genital warts up to *i*th participant;

*X*_1_ denotes sexual health, *X*_2_ represents the HPV factors (HPV vaccination, and HPV seminar), and *X*_3_ denotes demographic variables;

β_0_ is the model intercept, while *β*_1_, *β*_2_, and *β*_3_ denote the coefficients for sexual health and demographic characteristics;

$\varepsilon_{i}$ is the error term which was assumed to follow the binomial distribution.

Stata MP version 15 was used for statistical analysis, and *p*-values < 0.05 were considered significant at a 95% confidence interval (CI).

### 3.3. Ethical Considerations

The ethical approval for this study was obtained from the Niger State Ministry of Health Review Committee (protocol number: ERC PIN/2022/08/17 and approval number: ERC PAN/2022/08/17) in Minna, Niger State, Nigeria. Participants provided consent by clicking the “accept to participate” button after reading the informed consent statement on the first page. In order to assure the privacy and confidentiality of the participants, all participants voluntarily participated in this study and no identifying information was captured. Since data were collected online, the study has no/minimal risk to the health of the participants, their environment, and their relatives.

## 4. Results

### 4.1. Demographic Characteristics of the Study

The results in Table 1 below reveal the demographic characteristics of the students, non-academic staff, and academic staff who participated in this study (*n* = 1860). The participants’ age ranged between 15–64 years with Mean 32.1 ± 7.13 SD. The study contained more female than male participants, whereby 1750 (94.1%) of them were females and 110 (5.9%) were males. Most of the participants were millennials; thus, 1512 (81.3%) were aged 26–40 years, while 151 (8.1%) and 197 (10.6%) were aged 15–25 years and >40 years, respectively. Only 136 (7.3%) had a high school education and 1724 (92.7%) had a BSc or MSc or Doctoral Degree. Most of the participants recorded having an average livelihood, where 739 (39.7%) earned below 100 USD monthly, 27.4% earned between 100–200 USD, and 32.9% earned above 200 USD monthly. The majority of the population of 1369 (73.6%) were in a marital, sexually active relationship, and 26.4% were not in a marital, sexually active relationship.

### 4.2. Information about HPV and Sexual Health of Participants

The results in Table 2 below present the information about the participants’ self-reported HPV infection and sexual health. Mostly, 1843 (99.1%) were sexually exposed, where about a half (49.4%) had an early sexual debut from the ages of 15–20 years, 875 (47.5%) experienced a sexual debut between the ages of 21–30 years, and 57 (3.1%) had a sexual debut at age ≥31 years. It was observed that 569 (30.9%) participants have had only one sexual partner in a lifetime, 636 (34.5%) have had 2–4 sexual partners, and 638 (34.6%) have had ≥4 sexual partners. Among those who had more than one sexual partner, 934 (73.3%) practiced unprotected sex and 340 (26.7%) practiced protected sex, 1790 (97.1%) practiced vaginal sexual intercourse, and 53 (2.9%) practiced other (anal, oral, and vaginal) sexual intercourse. Furthermore, only 528 (28.4%) have attended an HPV seminar. Moreover, the rate of HPV vaccination was low, where just 107 (5.8%) among the participants have been vaccinated against HPV infection. Finally, 74 (4.0%) of the participants reported that they have had medical records of HPV genital warts, while 55 (3.0%) have had medical records of HPV-16 and -18 genotype infections.

### 4.3. The Participants’ Knowledge about the Association between Early Sex and HPV-16 and -18 Genotypes

Figure 2 below shows the population’s level of knowledge about the association between early sex and the HPV-16 and -18 genotypes. Quite a number of the participants agreed that early sex is associated with HPV infection, which means that 822 (44.2%) have knowledge about early sex, but knowledge about the HPV-16 and -18 genotypes was low as the statistical analysis revealed that only 149 (8.0%) among the participants agreed that they have any knowledge about the HPV-16 and -18 genotypes.

### 4.4. Association between HPV Infection and Participants’ Profile

At the time of the survey distribution, the results in Table 3 below show the association between participants who recorded self-reported HPV infection and the participants’ profile using chi-square test. The rate of HPV infection was 13 (8.6%) among those aged 15–25 years compared to the 56 (3.7%) and 5 (2.5%) rate among participants aged between 26–40 years and 41–64 years, respectively; *p* = 0.007. There was a preponderance of 27 (5.5%) of HPV infection among those who were not in a marital, sexually active relationship compared to married participants; *p* = 0.045. Similarly, more 48 (5.1%) of those who practiced unprotected sexual intercourse had HPV infection compared to those who practiced protected sexual intercourse; *p* = 0.096. Moreover, HPV infection was more common—5 (9.4%)—among those who practiced other (anal, oral, and vaginal) kinds of sex compared to those who practiced vaginal sex only; *p* = 0.083. HPV infection was 14 (9.4%) among those who have knowledge about the HPV-16 and -18 genotype compared to those—60 (3.5%)—who do not; *p* < 0.001. The proportion of HPV infection was slightly higher—42 (5.1%)—among participants who had knowledge about early sex compared to those who did not have knowledge about early sex; *p* = 0.027.

### 4.5. Factors Associated with HPV Genital Warts Infection among Students, Non-Academic Staff, and Academic Staff in North-Central Nigeria

The results in Table 4 represent the factors associated with HPV infection among the study participants using the logistic regression model. Only two variables had a significant association with HPV infection. The likelihood of HPV infection was lower among those aged between 26–40 years (_Adj_OR = 0.45, CI: 0.22–0.89, *p* = 0.024) and 41–64 years (_Adj_OR = 0.31, CI: 0.10–0.95, *p* = 0.041) compared to those aged 15–25 years.

## 5. Discussion

This study examined 1860 students, non-academic staff, and academic staff in north-central Nigeria on their demographic characteristics, their awareness/knowledge about HPV infections and HPV genotypes, and their sexual health and practices. The findings observed are a concern to public health.

Considering the demographic characteristics, we observed that the study sample was representative as it includes both high school/equivalent (low-levelled) and BSc, MSc, and Doctorate Degree (high-levelled) elite participants between the ages of 15–64 years, whereas most were between the age of 26–40 years. However, the gender distribution was not even as there were fewer males than females in the study. However, the survey also included some questions about cervical cancer, and this might be why male participants were not interested in taking the survey. The distribution of the participants based on their monthly income was fairly even. The distribution of the participants’ based on their marital status was skewed, where most of the participants indicated that they were in a sexually active relationship (married or separated), which is because the rate of divorce among ever married couples is low in the north-central region of Nigeria as compared to other regions [29]. This skewedness is expected because the north-central part of Nigeria is mostly populated by Muslims who believe that their religion detests divorce, according to Sunan Abu-Dawud, Book 6:2173 [30]. Moreover, a study conducted to examine cancer of the cervix in Zaria [31] and Lagos [32] revealed that most of the study participants were married which validates our result for having more participants who are currently in a marriage.

The result of the participants’ sexual behavior shows that a large majority of the participants reported having had sexual intercourse in their lifetime, and having their first experience between the age of 15–20 years, and more having had >4 partners in a lifetime. These results corroborate with other studies’ results that have explored the predictors of HPV among the male and female population, where early sex (≤20 years) is common among early and late adolescents in Nigeria [33], having 1 or more sexual partners before the age of 20 years [15], and having >4 sexual partners in a lifetime [34]. Furthermore, a vast number of the participants were also not concerned about practicing safe sex before they entered into a sexually active relationship. We excluded those who indicated having had only one sexual partner and analyzed those who have had >1 sexual partners (1274 participants) in a lifetime; the result revealed that most of the participants (73.3%) who fall under these group do not practice safe sex (with the use of condoms). This aligns with results from other studies that have reported that condoms are not widely used among non-married people in Nigeria [35]. A survey distributed by NOIpolls, in alliance with the National Agency for the Control of AIDS (NACA) and AIDS HealthCare Foundation (AHF) in 2020 to mitigate the spread of HIV/AIDS, reveals that only 34% of Nigerians indicated that they use condoms during sexual intercourse [36]. This study also reveals that the combination of anal, oral, and vaginal sex was low among the population, as 97.1% participants mostly practiced vaginal sex only. This is because the population under study might be affected by religious beliefs and detest the act of oral and anal sex. Another study also revealed that vaginal sex (95.2%) was mostly practiced among adolescents in Nigeria who were examined for self-reported HIV transmission [37].

The proportion of the participants with awareness of HPV-related seminars and the HPV vaccine was low (28.4% and 5.8%, respectively), similar to the corresponding percentage from a study that revealed that only 9.0% have had knowledge about HPV and cervical cancer programs/seminars [38], while another study examined secondary school female teachers in Lagos who have had an HPV vaccine (2.2%) given to their teenage children [32].

The chi-square test was used to measure the association between self-reported HPV infection and the participants’ profile. A statistical analysis shows that there was a significant association between HPV infection and the participants aged between 26–40 years (Fisher’s exact, *p* = 0.007), not in a marital sexually active relationship (Fisher’s exact, *p* = 0.045), who had no knowledge about HPV-16 and -18 genotypes (Fisher’s exact, *p* < 0.001), and who had no knowledge about early sex (Fisher’s exact, *p* = 0.027). The result about the participants who have self-reported having HPV infection correlated with the results from other studies where there exists an association between the population under study aged ≤ 30 years (Fisher’s exact, *p* = 0.006) [39], those with an unstable marital status or are single with multiple sexual partners (Fisher’s exact, *p* = 0.022 and *p* = 0.001, respectively) [40], and lack of knowledge about HPV (Fisher’s exact, *p* < 0.0001) [41] as predictors of HPV infection.

A regression analysis also reveals the factors predicting HPV genital warts among the population under study. The result shows that participants between the ages of 26–64 years have reported that they were likely carriers of HPV genital warts: 26–40 years with _Adj_OR = 0.45, CI: 0.22–0.89; and 41–64 years with _Adj_OR = 0.31, CI: 0.10–0.95). Thus, the participants between the ages of 26–40 years have higher odds as compared to those between the ages of 41–64 years (*p*-value, *p* = 0.024 and *p* = 0.041, respectively). This result aligns with previous studies where ages lower than or equal to 30 years are more likely to be infected than those whose ages are over 45 years [15,39,42,43]. Conversely, the variables of gender, marital status, kind of sex practiced, protected sex, and knowledge about early sex were not significantly associated with HPV infection in this study.

### 5.1. Study Strength and Limitations

The availability of a large sample size is one of the strengths of this study and it is also the first study to be conducted in Lapai, Niger State. It was also an achievement to include the male population in this study. However, one of the limitations faced by the study is that we had an extremely low sample size of the male population as compared to the female population. Therefore, we acknowledge that the male population in this study was not well-represented. An additional significant limitation of this study is that we have conducted the study based on self-reported data only. Hence, some responses provided by the participants to sensitive questions may have been biased. Furthermore, during data collection, we did not provide categorization for students, non-academic staff, and academic staff; thus, the *Human Papillomavirus (HPV) and Cervical Cancer Risk Assessment (HCRA) Tool (questionnaire)* has been modified and currently used for the last phase of the research field work for data collection.

### 5.2. Conclusions

HPV infection was high among participants (who have self-reported having a medical/history of the HPV-16 and -18 genotype) between the ages of 26–40 years, currently not in a sexually active relationship, and who had no knowledge about the HPV-16/-18 genotypes and the consequences of an early sexual debut. Moreover, participants > 26 years of age are likely to be carriers of HPV genital warts. These factors have been revealed as the most important predictors peculiar to IBB university students, non-academic staff, and academic staff in this study. Hence, the need for awareness about HPV is required/a must for the safety and well-being of the IBB university community. As a concern and solution, this situation, therefore, suggests the need for free routine HPV vaccination and programs in universities in Nigeria, as well as stresses the importance of sex education among adults through awareness programs in the academic environment populated with low- and middle-income earners in Nigeria.

## Figures and Tables

**Figure 1 viruses-16-00902-f001:**
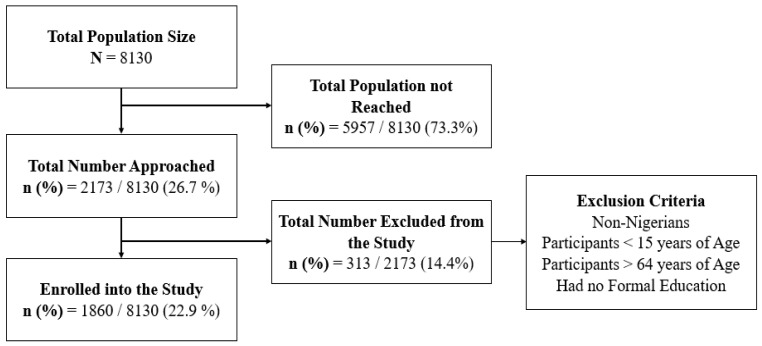
Study enrolment flowchart.

**Figure 2 viruses-16-00902-f002:**
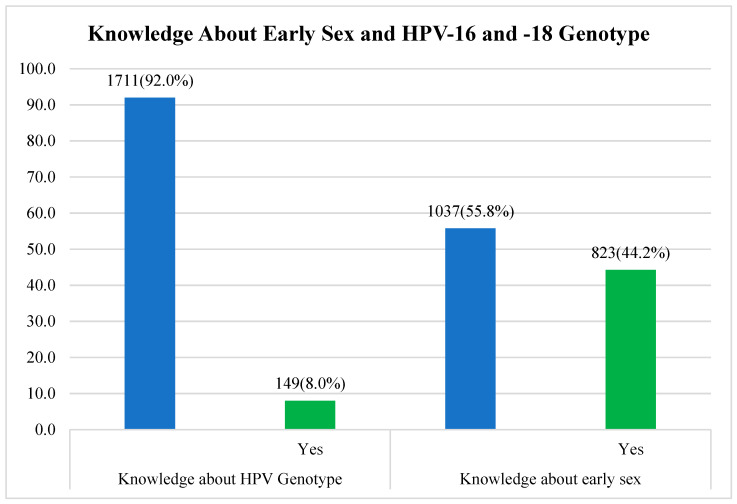
Knowledge about early sex and HPV-16 and -18 genotypes.

**Table 1 viruses-16-00902-t001:** Demographic characteristics of the participants.

Variables	Frequency (*n* = 1860)	Percent (%)
**Gender**		
Male	110	5.9
Female	1750	94.1
**Age**		
15–25	151	8.1
26–40	1512	81.3
41–64	197	10.6
**Education**		
High School/Equivalent	136	7.3
BSc, MSc, Doctorate Degree	1724	92.7
**Income (USD)**		
Below 100	739	39.7
100–200	509	27.4
Above 200	612	32.9
**Marital status**		
Currently married	1369	73.6
Not in a sexually active relationship (Single/Divorced)	491	26.4

**Table 2 viruses-16-00902-t002:** Sexual health and HPV information.

Variables	Frequency (*n* = 1860)	Percent (%)
**Have you ever had sex?**		
Yes	1843	99.1
No	17	0.9
**Age at first sexual intercourse (*n* = 1843)**		
15–20	911	49.4
21–30	875	47.5
31 and above	57	3.1
**Number of sexual partners in lifetime** **(*n* = 1843)**		
1 sexual partner in a lifetime	569	30.9
2–4 sexual partners in a lifetime	636	34.5
>4 sexual partners in a lifetime	638	34.6
**Protected sex (*n* = 1274)**		
Unprotected sex	934	73.3
Protected sex	340	26.7
**Kind of sex practiced (*n* = 1843)**		
Vaginal sex	1790	97.1
Others (anal, oral, vaginal)	53	2.9
**HPV seminar**		
No	1332	71.6
Yes	528	28.4
**Ever received HPV vaccine?**		
No	1753	94.2
Yes	107	5.8
**Medical record of HPV-16 and -18 genotypes**		
No	1805	97.0
Yes	55	3.0
**Medical record of HPV genital warts infection**		
No	1786	96.0
Yes	74	4.0

**Table 3 viruses-16-00902-t003:** Association between HPV infection (medical records of HPV infection) and the participants’ profile.

	Self-Reported HPV Infection			
Variables	No	Yes	Test Statistics (Chi-Square/Fisher’s Exact)	*p*-Value
**Gender**			**0.67**	**0.414**
Male	104(94.5)	6(5.5)		
Female	1682(96.1)	68(3.9)		
**Age**			**9.85**	**0.007 ****
15–25	138(91.4)	13(8.6)		
26–40	1456(96.3)	56(3.7)		
41–64	192(97.5)	5(2.5)		
**Education**			**0.04**	**0.852**
Elementary School and High School	131(96.3)	5(3.7)		
BSc, MSc, Doctorate Degree	1655(96.0)	69(4.0)		
**Income**			**1.41**	**0.495**
Below 100	706(95.5)	33(4.5)		
100–200	493(96.9)	16(3.1)		
Above 200	587(95.9)	25(4.1)		
**Marital status**			**4.07**	**0.045**
Married	1322(96.6)	47(3.4)		
Not in a sexually active relationship (single/divorced)	464(94.5)	27(5.5)		
**Have you ever had sexual intercourse?**			**0.71**	**0.645**
No	17(100.0)	0(0.0)		
Yes	1769(96.0)	74(4.0)		
**Age at first sexual intercourse**			**1.07**	**0.585**
15–20	876(96.2)	35(3.8)		
21–30	837(95.7)	38(4.3)		
≥31	56(98.2)	1(1.8)		
**Number of sexual partners in lifetime**			**3.4**	**0.182**
1 sexual partner in a lifetime	553(97.2)	16(2.8)		
2–4 sexual partners in a lifetime	609(95.8)	27(4.2)		
≥4 sexual partners in a lifetime	607(95.1)	31(4.9)		
**Protected sex**			**2.77**	**0.096 ***
Unprotected sex	886(94.9)	48(5.1)		
Protected sex	330(97.1)	10(2.9)		
**Kind of sex practiced**			**4.97**	**0.083 ***
Vaginal sex only	1721(96.1)	69(3.9)		
Others (anal, oral, vaginal)	48(90.6)	5(9.4)		
**HPV seminar**			**0.28**	**0.600**
No	1281(96.2)	51(3.8)		
Yes	505(95.6)	23(4.4)		
**Knowledge about HPV-16 and -18 genotype**			**12.44**	**0.000 ****
No	1651(96.5)	60(3.5)		
Yes	135(90.6)	14(9.4)		
**HPV vaccine?**			**1.95**	**0.162**
No	1686(96.2)	67(3.8)		
Yes	100(93.5)	7(6.5)		
**Knowledge about early sex**			**4.89**	**0.027 ****
No	1005(96.9)	32(3.1)		
Yes	781(94.9)	42(5.1)		

* *p*-value is significant at 10% level of significance. ** *p*-value is significant at 5% level of significance.

**Table 4 viruses-16-00902-t004:** Factors associated with HPV genital warts infection among students, non-academic staff, and academic staff in north-central Nigeria.

Self-Reported HPV Genital Warts	Odds Ratio	95% CI		*p*-Value	
		**Lower**		**Upper**	
**Gender**					
Male	*ref*				
Female	0.80	0.32		1.99	0.630
**Age**					
15–25	*ref*				
26–40	0.45	0.22		0.89	0.024
41–64	0.31	0.10		0.95	0.041
**Marital status**					
Married	*ref*				
Not in a sexually active relationship (Single/Divorced)	1.39	0.81		2.40	0.233
**Kind of sex practiced**					
Vaginal sex	*ref*				
Others (anal, oral, vaginal)	2.13	0.77		5.84	0.143
**Protected sex**					
Unprotected sex	*ref*				
Protected sex	0.59	0.32		1.09	0.095

## Data Availability

The HPV and CC data collected during this project and processed for this research publication as part of this study were submitted to the Niger State Ministry of Health Ethical Review Committee (NSMOH ERC) and are available upon request. Data is contained within the article.

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
