# Peer review of "Factors Associated with HPV Genital Warts: A Self-Reported Cross-Sectional Study among Students and Staff of a Northern University in Nigeria"

_viruses, 2024, doi:10.3390/v16060902_

Round 1

Reviewer 1 Report

Comments and Suggestions for Authors

Overall, the study is very good performed and written. It is very comprehensive and gives important data. I suggest to accept it with minor changes listed bellow.

Abstract:

The sentence … “HPV is one of the most common Sexually Transmitted Infections (STIs) in Nigeria with high-risk HPV-16 and 18 genotypes.” … is not adequately written.

Introduction:

The sentence … “However, North-central Nigerian adults are faced with some challenges which have contributed to the spread of HPV infection as against the unavailability of some protective measures due to the lack of knowledge about HPV in general, abject poverty (resulting into inability to: purchase HPV vaccines, eat healthy food, and attend paid HPV-related seminars), individual’s ethical imperatives, unavailability of free medical services, and some National policies that are not in place to make HPV vaccine available at a low cost and enforce it as a part of the routine vaccination program [13]…. is too long.

Results:

Page 9, line 266: You probably meant on age 15-25.

Table 4: the same comment as above (age 15-25).

Comments on the Quality of English Language

English Language is fine. Only minor editing is needed.

Author Response

Reviewer 1

Comments and Suggestions for Authors

Overall, the study is very good performed and written. It is very comprehensive and gives important data. I suggest to accept it with minor changes listed bellow.

Abstract:

The sentence … “HPV is one of the most common Sexually Transmitted Infections (STIs) in Nigeria with high-risk HPV-16 and 18 genotypes.” … is not adequately written.

Sentence revised.

Introduction:

The sentence … “However, North-central Nigerian adults are faced with some challenges which have contributed to the spread of HPV infection as against the unavailability of some protective measures due to the lack of knowledge about HPV in general, abject poverty (resulting into inability to: purchase HPV vaccines, eat healthy food, and attend paid HPV-related seminars), individual’s ethical imperatives, unavailability of free medical services, and some National policies that are not in place to make HPV vaccine available at a low cost and enforce it as a part of the routine vaccination program [13]…. is too long.

Sentence simplified and broken into smaller sentences.

Results:

Page 9, line 266: You probably meant on age 15-25.

Yes, corrected.

Table 4: the same comment as above (age 15-25).

Yes, it is the same.

Reviewer 2 Report

Comments and Suggestions for Authors

Well designed study and manuscript.

1. The affiliation of author number 6 is incomplete. Please provide full affiliation.

2. What is the role of the authors from Budapest or other countries in this study? What is their contribution, especially the 6th author, given as Medical Devices

3. Union may mean sexual activity for you. Kindly change it to - a sexually active relationship.

4. It is unclear as to why you wish the vaccination to be given to only students of your own university. This appears to be a vested interest. Indeed, the correct age group is 9 to 14 years. 

5. Why were students between 15 and 25 years in the University? What course were such students pursuing?

6. The study seems to be disproportionately representing male population: a mere 5%. Is that representative of the university statistics?

7. Can students at the University can be mapped to various regions/districts/states of Nigeria or, are they a homogenous population from 1 single region? If they are from different regions, a representative map can be included specifying the various regions these students are from

8. A question that could have been included in the google form could be: were you made aware of teenage sex, unprotected sex in your high school. 

9. It is assumed that every student in the university had a face book account into which a link to the google form was posted. If not, were only students with a face book  account included in the study? 

10 It is also an assumption that every single student had a smartphone, a tab or a laptop to access the questionnaire. Is this correct?

11. Apart from universal vaccination being recommended, what other measures would you like to propose to reduce HPV induced genital warts?

Comments on the Quality of English Language

Need to be reviewed for quaint expressions

Author Response

Reviewer 2

Comments and Suggestions for Authors

Well designed study and manuscript.

  1. The affiliation of author number 6 is incomplete. Please provide full affiliation.

Thank you for your comprehensive review of this manuscript; your input is greatly appreciated as it enhances the quality of the document.

Full affiliation added.

  1. What is the role of the authors from Budapest or other countries in this study? What is their contribution, especially the 6th author, given as Medical Devices

The role and contribution of authors includes the following;

  • Melvin Omone Ogbolu – Bioinformatics (Biostatistics and Biomedical Engineering) / Owner of PhD research project, Drafting of the manuscript and data analysis.
  • Olanrewaju D. Eniade – Epidemiology and Medical Statistics / Drafting of the manuscript and data analysis.
  • Dr. Hussaini Majiya – Microbiologist and Biotechnology / Advise on drafting manuscript and seminar organization for awareness creation of HPV and data collection in IBB University in Nigeria.
  • Prof. Miklós Kozlovszky – Bioinformatics (Telemedicine and Biomedical Engineering) / First author’s PhD project supervisor and adviser for all projects and manuscripts (reviewed and editted the manuscript, advise for analytical methods).
  1. Union may mean sexual activity for you. Kindly change it to - a sexually active relationship.

Suggest added.

  1. It is unclear as to why you wish the vaccination to be given to only students of your own university. This appears to be a vested interest. Indeed, the correct age group is 9 to 14 years. 

In section 5.2 – Conclusions, 364 – 368, we have indicated that HPV vaccination should be administered across Universities in Nigeria.

As a concern and solution, this situation can therefore suggest the need for free HPV vaccination routine and programs in universities in Nigeria, as well as to stress the im-portance of sex education among adults through awareness programs in the academic environment populated with low- and middle-income earners in Nigeria.

  1. Why were students between 15 and 25 years in the University? What course were such students pursuing?

There are several courses registered at the university. Thus, we didn’t approach the students based on their course of study. In Nigeria, there are younger (between 13-15) and older (above 22-25) students in the tertiary institutions.

  1. The study seems to be disproportionately representing the male population: a mere 5%. Is that representative of the university statistics?

In general, there are fewer males who would be interested in responding to questionnaires. In such a case, this is the reason for the fewer responses of males in this study. Furthermore, our research topic is mostly directed towards women.

In section 5.1, we have also included this as a limitation in this study.

It was also an achievement to include the male population in this study. However, one of the limitations faced by the study is that we had an extremely low sample size of the male population as compared to the female population.

  1. Can students at the University can be mapped to various regions/districts/states of Nigeria or, are they a homogenous population from 1 single region? If they are from different regions, a representative map can be included specifying the various regions these students are from

IBB University comprises of students and staff from various regions in Nigeria, which includes Igbo, Yoruba, and Hausa. However, data to determine which regions the participants are from were not captured in our questionnaire.

  1. A question that could have been included in the google form could be: were you made aware of teenage sex, unprotected sex in your high school. 

Thank you very much for this suggestion but we didn’t include this kind of question. However, we have the following questions which are similar to the suggestion in our questionnaire;

Q11. Have you ever attended any seminars or training about HPV and HPV Vaccine or read about HPV and HPV Vaccine?

Yes

No

Q15. Do you know that having sex at an early age increases the risk of getting HPV?

Yes

No

ETC.

  1. It is assumed that every student in the university had a face book account into which a link to the google form was posted. If not, were only students with a face book  account included in the study? 

No, not all the students have a Facebook account. Some of the students answered the questionnaire via email. This has been added to the manuscript.

10 It is also an assumption that every single student had a smartphone, a tab or a laptop to access the questionnaire. Is this correct?

Yes

  1. Apart from universal vaccination being recommended, what other measures would you like to propose to reduce HPV induced genital warts?

The following measures would be suggested;

  1. Safe sex practices: such as the use of condoms.
  2. Healthy diet: food high in vitamin C and antioxidants.
  3. Frequent medical checkups: Can help to reduce the risk of persistent HPV infections.
  4. Limited number of sexual partners in one's lifetime.

Comments on the Quality of English Language

Need to be reviewed for quaint expressions

Revised